# Global Trends in Research on Cell-Free Nucleic Acids in Obstetrics and Gynecology during 2017–2021

**DOI:** 10.3390/jcm11195545

**Published:** 2022-09-22

**Authors:** Wenyan Gao, Hongyue Yang, Wanting Cheng, Xiao Wang, Da Li, Bei Shi

**Affiliations:** 1Department of Obstetrics, The First Affiliated Hospital of China Medical University, Shenyang 110001, China; 2Center of Reproductive Medicine, Shengjing Hospital of China Medical University, Shenyang 110004, China; 3Department of Physiology, School of Life Sciences, China Medical University, Shenyang 110122, China; 4Key Laboratory of Reproductive Dysfunction Diseases and Fertility Remodeling of Liaoning Province, Shenyang 110122, China

**Keywords:** serum screening, genetics, perinatal diagnosis invasive, prenatal screening

## Abstract

Objectives. The objectives of this study were to identify global trends in research on cell-free deoxyribonucleic acid (cfDNA) from a bibliometric perspective and provide researchers with new research hotspots. Methods. In all, we extracted 5038 pieces of literature from PubMed and 527 articles from the Web of Science Core Collection (WoSCC) database related to cfDNA published from 1 January 2017 to 31 December 2021. For PubMed literature, we employed co-word, biclustering, and strategic diagram analysis to describe the trends in research on cfDNA in the said five years. Then, we used VOSviewer analysis for the WoSCC database to display the trends in research on cfDNA in obstetrics and gynecology during 2017–2021. Results. Strategy diagram analysis of 95 major Medical Subject Headings terms extracted from 5038 pieces of literature indicated that cfDNA sequence analysis for non-invasive prenatal and genetic testing and its application in the fields of neoplasm genetics and diagnosis is a newly emerging immature theme of cfDNA. VOSviewer analysis of 527 articles showed the global trends in research on cfDNA in obstetrics and gynecology, for example, in terms of most influential authors, institutions, countries, journals, and five research hotspots: (1) cfDNA application in prenatal screening and prenatal diagnosis, (2) cfDNA application in assisted reproductive technology, (3) cfDNA application in pre-eclampsia, DNA methylation, etc., (4) cfDNA application in placental dysfunction and fetal growth restriction, and (5) cfDNA application in fetal chromosomal abnormalities (fetal aneuploidy). Conclusions. Comprehensive visual analysis provides information regarding authors, organizations, countries/regions, journals, research hotspots, and emerging topics in the field of cfDNA for obstetrics and gynecology research. This comprehensive study could make it easier to find a partner for project development and build a network of knowledge on this emerging topic.

## 1. Introduction

Cell-free nucleic acids (cfNAs) are found outside the cells/nuclei, in the plasma, and in various other biological fluids, including DNA, mRNA, mitochondrial DNA, etc. Cell-free deoxyribonucleic acid (cfDNA) exhibits high stability in body fluids and has all the characteristics of nucleic acid in the cell of its origin, making it a suitable candidate biomarker that can be detected for clinical applications (an emerging field in non-invasive evaluation), such as prenatal diagnosis, cancer, diabetes, cardiovascular, and neurological diseases [1,2,3].

However, so far, there has been no systematic analysis of cfDNA to identify its research trends. In this study, we (1) searched the subject terms of cfDNA through literature by performing bibliometric analysis, (2) analyzed the current status and trends in cfDNA research, and (3) focused on its application in the field of obstetrics and gynecology. We also explored the time evolution of research keywords in different topics all to identify the research hotspots and emerging themes of cfDNA in the field of obstetrics and gynecology.

## 2. Methods

### 2.1. Data Source and Research Process

The PubMed database was searched as the source of this research. The Medical Subject Headings (MeSH) phrase “cell-free nucleic acids” was used, and the time span was “from 1 January 2017 to 31 December 2021”. The language was English. This part of the study was performed to describe the trends in research on cfDNA in the selected five years. Two independent investigators screened these articles. The literature was excluded if it was not related to cfDNA study or if the basic information was incomplete. Finally, 5038 articles were chosen (Figure 1).The Web of Science Core Collection (WoSCC) database was searched to identify the trends in research on cfDNA in obstetrics and gynecology. The phrases “cell-free DNA” or “cell-free nucleic acids” were used for the search, and the time span was “from 2017 to 2021”. We refined the categories of the retrieved documents and limited the documents to those related to obstetrics and gynecology, developmental biology, and reproductive biology. The retrieved results were saved as “plain text” and “full record and cited references”. We collected the basic information about each article, such as country, author, institution, journal, references, and keywords. Two investigators independently screened the results of the primary search, and literature not related to cfDNA or missing basic information was excluded. Finally, 527 articles were chosen (Figure 1).

### 2.2. Analysis Tools and Methods

To extract the relevant information on cfDNA from the literature downloaded from PubMed, including publication year, author, journal, country, and major MeSH terms/MeSH subheadings, and determine the distribution characteristics, we used the co-occurrence matrix generator the Bibliographic Item Co-Occurrence Matrix Builder (BICOMB) [4]. To cluster these high-frequency MeSH terms/MeSH subheadings and visualize the mountain and the matrix, we used the threshold (T) to determine the high-frequency MeSH terms/MeSH subheadings [5] and gCLUTO software (http://glaros.dtc.umn.edu/gkhome/cluto/gcluto, accessed on 21 February 2022). On the basis of the co-occurrence matrix, we calculated the centrality and density of each theme cluster. According to the theme density and the centrality, we built a 2D strategic diagram by plotting themes along two axes. The x-axis represented the centrality or the external cohesion index, i.e., the central position of the theme within the overall network. The y-axis represented the density or the internal cohesion index, i.e., the conceptual development of the theme. Thus, the x- and y-axes generated four quadrants. Graph Pad 5 software was used to generate the strategic diagram. All the above analyses processes were the same as described in our previous study [6].

In the third quadrant of the strategic diagram, we conducted VOSviewer analysis and research on topics related to obstetrics and gynecology. VOSviewer enables visualization and displays cluster analysis results. It can support multiple types of bibliometric research. To analyze the authors, institutions, countries, journals, and keywords of the literature and build a visual map of keywords, we used VOSviewer version 1.6.13.

## 3. Results

### 3.1. Research Hotspots Identified on the Basis of MeSH Term Theme Clusters

From the 5038 articles extracted from the PubMed database, we retrieved a total of 95 high-frequency MeSH terms/MeSH subheadings and sorted them in descending order of frequency (Appendix A). Then, we performed biclustering analysis of these high-frequency MeSH terms/MeSH subheadings to form six clusters (Appendix A). The analyses results are presented in the form of mountain peaks and matrix visualization in Figure 2A,B. The top of the cluster trees shows the relationship between MeSH keywords and articles. Appendix A displays 95 MeSH terms/MeSH subheadings and the relationship between the MeSH terms and their clusters.

### 3.2. cfDNA Theme Trends

According to the biclustering analysis of MeSH terms/MeSH subheadings from the PubMed database, the main MeSH terms/MeSH subheadings of cfDNA were assigned to the four quadrants in the strategic diagram (Figure 2C,D).

Motor themes are those with both strong centrality and high density, as shown in Quadrant I (upper right). Specialized themes are in Quadrant II (upper left) and are defined as those with inadequate external interactions but high density. Quadrant III (lower left) contains themes with weak density and inadequate centrality, and these themes are usually considered to be either appearing or vanishing. The last quadrant, Quadrant IV (lower right), contains themes with strong centrality but a lack of internal maturation [7].

Themes in Quadrant I of the strategic diagram are considered to be the central themes in this research field. The main research areas involved are cfDNA detection methods and application in neoplasm genetics, including DNA methylation and mutation analysis methods, lung neoplasm drug therapy and antineoplastic agents, early detection of cancer, and colorectal and pancreatic neoplasm genetics. Quadrant III contains peripheral and immature themes, which means the research objects are not yet a research hotspot the studies have small samples, or the experimental results need to be further confirmed. The research on cfDNA in this field mainly includes its application in non-invasive prenatal testing, neoplasm metabolism, and neoplasm diagnosis, including cell-free nucleic acid sequence analysis for non-invasive prenatal testing and genetic testing, cell-free nucleic acid isolation and purification, circulating neoplastic cell metabolism, and genetics and diagnosis for neoplasms. Quadrant IV contains the central themes of immature research and mainly refers to the application of cfDNA in precision medicine, including circulating tumor RNA metabolism and analysis.

### 3.3. cfDNA Application in Obstetrics and Gynecology

A total of 527 articles were retrieved from WoSCC, and information related to the author (intercepting frequency of 3), author institution (intercepting frequency of 5), country (intercepting frequency of 3), and journals (intercepting frequency of 3) was extracted. The following is an explanation of the results of our search.

### 3.4. Most Influential Authors, Institutions, Countries, and Journals in cfDNA Research on Obstetrics and Gynecology

We listed the top 10 authors, institutions, countries, and journals that had published the most articles related to cfDNA. The authors with the most published articles are Dugoff Lorraine, Norton Mary E, and Caughey Aaron B (Figure 3A; Table 1). The most influential institutions are mostly located in Australia. Among them, the influence of Univ Melbourne ranks first (Figure 3B), with a total of 181 citations (Table 2). The countries that have published the maximum number of articles are the United States, China, and England (Figure 3C). The number of publications was 189, 82, and 42, and the citations were 1012, 255, and 378, respectively (Table 3). Prenatal Diagnosis is the journal that has published the maximum number of articles in this field (Figure 3D), with a total of 90 articles, followed by Ultrasound in Obstetrics & Gynecology and the American Journal of Obstetrics and Gynecology, with 51 and 48 articles, respectively. Most of these influential journals were founded in England and the United States (Table 4).

### 3.5. Most Influential Keywords in cfDNA Research in Obstetrics and Gynecology

The keywords extracted from the 527 articles were clustered into five hot topics (Figure 4A; Table 5). Next, we identified the trends in the years of the appearance of keywords from 2017 to 2021 (Figure 4B), for example, non-invasive prenatal testing appeared around 2021. In 2017, the main research hotspots focused on the clinical application of prenatal screening, cell-free DNA, genetic testing, and trisomy 13. In recent years, cfDNA has made further progress in the fields of trophectoderm biopsy, preimplantation genetic testing, DNA methylation, etc. We generated a high-frequency keyword density map of cfDNA (Figure 4C). It is clear that bright-yellow entries, such as prenatal diagnosis, prenatal screening, cell-free fetal DNA, and fetal fraction currently occupy a leading position in research and have great significance in this field.

## 4. Discussion

cfDNA was first discovered by Mandel and Metais in 1947 [8], but due to the lack of highly sensitive and specific experimental methods, the research on the relationship between cfDNA and disease has progressed slowly for a long time. With the emergence of effective separation technology of cfDNA, and the application of detection technology combined with special fluorescent dyes and PCR technology, in recent years, the research in this field has developed rapidly. This article mainly discusses the value and prospects of cfDNA in the field of obstetrics and gynecology.

### 4.1. Crucial Themes in the Strategy Diagram

Quadrant I in the strategy diagram includes subjects of mature research in the field of cfDNA research, such as monitoring neoplastic cell metabolism, detecting early cancer, and applications of cfDNA in DNA methylation and mutation analysis methods. In tumor patients, most cfDNA is released through tumor cell apoptosis or necrosis [9]. cfDNA can be collected from the body fluids of patients multiple times non-invasively, thus overcoming the limitations of a single biopsy. In addition, we can take samples during different treatment periods to monitor the patient’s response to tumor treatment. This is an exciting step in the transformation of cfDNA research into clinical application, and it is worthy of further research [10]. The protein kinase family has emerged as one of the most important drug targets of the 21st century due to dysregulation of protein kinase activity in many diseases, including cancer. In 2021, the FDA approved a variety of protein kinase inhibitors for the treatment of malignant tumors, such as lung neoplasms and colorectal neoplasms [11,12].

### 4.2. Peripheral and Undeveloped Themes in the Strategy Diagram

The research hotspots in Quadrant III mainly include cell-free nucleic acid sequence analysis for non-invasive prenatal testing; cell-free nucleic acid analysis for genetic testing; and the application of cfDNA in the fields of neoplasm genetics and diagnosis, coronary artery disease, and type 2 diabetes (Figure 5). For chromosome disorders, such as balanced translocation, mosaicism results in a poor outcome in assisted reproductive technology (ART). Compared with the traditional TE biopsy technology, the non-invasive detection technology of cfDNA is playing an increasingly important role in PGT and genetic detection. Many studies have been devoted to promoting its application in preimplantation embryo screening and ART [13,14]. The non-invasive detection technology of cfDNA is considered to have bright prospects in ART. Liquid biopsies based on plasma cell-free DNA analysis can provide diagnostic information, and this technology is currently used in the targeted therapy of neoplasms and monitoring the disease progression. In future studies, the alignment of DNA methylation, fragmentome, and topological analysis of cell-free DNA in a targeted or genome-wide manner is expected to have a beneficial impact on clinical practice [15]. In diseases with metabolic disorders, such as type 2 diabetes [16,17] and coronary artery disease [18], there are also corresponding cells, such as pancreatic β-cell, whose cell-free DNA fragments are modified, making these cells potential biomarkers of the diseases.

### 4.3. Central and Immature Themes in the Strategy Diagram

Quadrant IV includes the central themes of immature research, mainly precision medicine and circulating tumor RNA metabolism and analysis. Different epigenetic modifications of cfDNA can show disease-related specificity. Therefore, cfDNA is the latest development in precision medicine in clinically relevant diagnosis and prediction. For example, cfDNA is extracted for molecular tumor analysis and detection of DNA methylation and is used widely in personalized clinical diagnosis [19] and clinical pathology [20]. As an immature central theme, precision medicine should be further studied.

### 4.4. Global cfDNA Trends in Obstetrics and Gynecology

We searched the WOSCC database for the application of cfDNA in the field of obstetrics and gynecology. We found a high-quality collaboration among authors publishing cfDNA-related articles. Dugoff Lorraine is the author who has published the maximum number of articles, and Levy Brynn is a co-cited author in 119 articles. In addition, among the top 10 authors, Dugoff Lorraine and Hui Lisa as well as Menezes Melody and Chitty Lyn S. have extensive partnerships. Among the top 10 countries, the United States, China, and England rank among the top three, accounting for 61.74% of the total (313/507). They have become the main forces driving this research forward. However, Chinese articles have only been cited 255 times, which is significantly lower than articles from the United States (1012 citations) and England (378 citations). This shows that the quality of Chinese articles in this research field needs to be further improved. Among the top 10 journals, 4 journals are from England, 3 are from the United States, 2 are from Switzerland, and 1 journal is from the Netherlands. The top three journals (*Prenatal Diagnosis*, *Ultrasound in Obstetrics & Gynecology*, and *American Journal of Obstetrics and Gynecology*) are influential in the field of obstetrics and gynecology, which fully demonstrates that cfDNA is important for research in the field of obstetrics and gynecology.

### 4.5. Knowledge Base, Hotspots, and Emerging Frontiers Related to cfDNA in Obstetrics and Gynecology

Then, we searched the WOS database for “cell-free nucleic acids” or “cell-free DNA” and refined the category to the literature on obstetrics and gynecology from 2017 to 2021, focusing on the study of cfDNA in obstetrics and gynecology, especially application in the field of obstetrics (Figure 6). The co-occurrence analysis of the top 46 keywords of cfDNA divided into five hot topics helps focus on the next research direction.

Cluster 1 (Figure 4A, red cluster): cfDNA application in prenatal screening and prenatal diagnosis.

In the red cluster, the more frequently occurring keywords are prenatal diagnosis, prenatal screening, amniocentesis, chorionic villus sampling, etc. In recent years, non-invasive sampling (liquid biopsy) has become a popular method for prenatal screening and diagnosis and new molecular biology techniques have been developed. Many professional associations, including the Society of Maternal-Fetal Medicine (SMFM) and the American Congress of Obstetricians and Gynecologists (ACOG), have issued recommendations on the proper use of cfDNA screening during pregnancy [21,22,23]. With the recent increase in the adoption of cfDNA screening, the frequency of application of chorionic villus sampling (CVS) and amniocentesis has decreased significantly, but they remain important prenatal diagnostic methods in the first and second trimesters, and amniocentesis analysis is still the only method to see balanced rearrangement and mosaicism. In addition to assessing genetic diseases, amniocentesis can be used to assess the presence of intra-amniotic or fetal infections by culture or polymerase chain reaction or to assess neural tube defects by measuring alpha-fetoprotein and acetylcholinesterase in amniotic fluid. The increasing availability and application of microarray cytogenetic [24] testing has led to the discovery of an increasing number of chromosomal abnormalities of unknown clinical significance. In this case, parental studies are usually considered to determine whether the variation exists in any parent and to evaluate the possible clinical significance of the variation.

Cluster 2 (Figure 4A, green cluster): cfDNA application in assisted reproductive technology.

In the green cluster, aneuploidy, preimplantation genetic testing (PGT) and screening, in vitro fertilization (IVF), preimplantation genetic testing-aneuploidy (PGT-A), etc. appear more frequently. The clinical application of non-invasive preimplantation genetic testing (niPGT), a non-invasive method of PGT, has demonstrated the potential reliability of cfDNA as a genetic evaluation resource. Mosaicism, as an individual composed of cell lines from different zygotes, may limit the accuracy of prenatal screening and prenatal diagnosis of cfDNA. Currently, Zhou’s team [25] is studying the specificity of niPGT under different chimera threshold conditions and found that after adjusting the mosaic threshold, the specificity of niPGT increased from 69.7% to 84.8% in terms of overall ploidy and from 96.1% to 98.9% at the chromosome level. This increases the possibility of using cfDNA for genetic evaluation. It is worth mentioning that with the improvement of next-generation sequencing (NGS) technology, cfDNA is being increasingly applied in preimplantation genetic testing (PGT). By using the SurePlex WGA method, sufficient cfDNA can be extracted from blastocoel fluid and spent blastocyst medium for whole-genome amplification (WGA) and accurate aneuploidy screening is achieved [26]. Jiao et al. reported that a new assay called minimally invasive chromosome screening (MICS) can detect comprehensive chromosome ploidy with a high resolution by using cfDNA extracted from the blastocyst culture medium (BCM) [27]. Similarly, mosaicism detected by a single biopsy of trophectoderm (TE) cells shows poor reliability and accuracy. Recently, a pilot study of niPGT of cfDNA in blastocyst culture medium exhibited an effective diagnostic performance in putative mosaic blastocysts. Therefore, the number of blastocysts available for transfer may increase and the overall clinical outcomes may improve by performing inPGT of cfDNA in blastocyst culture medium [25].

Cluster 3 (Figure 4A, blue cluster): cfDNA application in pre-eclampsia.

In the blue cluster, pre-eclampsia, toll-like receptor 9, DNA methylation, etc., appear more frequently. Pre-eclampsia is a serious complication during pregnancy. Since the pathogenesis of pre-eclampsia cannot be explained by “monism,” so far, there is no single parameter that can predict the occurrence and development of pre-eclampsia. In 1999, Lo et al. [28] found a five-fold increase in cfDNA levels in a small sample of 20 women with pre-eclampsia compared to those in normotensive pregnant women. More importantly, the elevation of cfDNA levels predates the onset of clinical symptoms. Later studies by Leung [29], Zhong [30], and Levine [31] also confirmed that the level of cfDNA in women with pre-eclampsia was significantly higher than that in women with a normal pregnancy. Notably, in women with mild pre-eclampsia, cfDNA levels were similar to those in women with a normal pregnancy [32]. Scharfe-Nugent et al. proposed that cfDNA has a pro-inflammatory effect, which was demonstrated in BALB/c mice: interleukin (IL)-6 production was detected after intraperitoneal administration of cfDNA. Furthermore, this effect appears to be dependent on toll-like receptor 9 (TLR-9), as TLR-9(-/-) mice were shown to be immune to inflammation induction [33]. This finding may shed light on the underlying mechanism by which cfDNA promotes the pathophysiological changes in pre-eclampsia.

Cluster 4 (Figure 4A, yellow cluster): cfDNA application in fetal growth restriction and placental dysfunction.

In the yellow cluster, fetal fraction, fetal growth restriction (FGR), placenta, placental dysfunction, etc., appear more frequently and are related to pregnancy complications. At present, maternal serum cell-free DNA analysis has been used clinically to screen for genetic abnormalities during pregnancy, but there is limited work on its use in detecting placental function. Some scholars have conducted research on whether the detection of maternal serum cfDNA during pregnancy can help predict placental dysfunction [34] or adverse pregnancy outcomes (APOs) [35] and found that some specific cfDNA could predict pre-eclampsia or FGR [36] but cannot predict gestational diabetes [37]. It means that relying on a single cfDNA strategy may not be sufficient to accurately predict different APOs. This new predictive study currently has a small sample size, but it adopts a longitudinally designed prospective cohort study method to provide feasibility for future larger-scale, multi-center clinical trials [35]. It is worth mentioning that the concentration of mitochondrial-DNA (mtDNA) changes with the gestational week during pregnancy. Compared with early pregnancy and the postpartum period, the serum-free mtDNA concentration in late pregnancy is higher. The concentration of circulating mtDNA in a normal pregnancy can be used as a reference value for clinical prognosis or diagnostic testing of pregnant women who have or are at risk of pregnancy complications [38].

Cluster 5 (Figure 4A, purple cluster): cfDNA application in fetal chromosomal abnormalities (fetal aneuploidy)

In the last cluster, the purple cluster, trisomy 21, trisomy 13, trisomy 18, twin pregnancy, etc., appear more frequently. When ultrasound indicates increased nuchal translucency in the fetus in the first trimester, it is highly suspected that the fetus has chromosomal aneuploidy abnormalities, such as trisomy 21. Prenatal screening of cfDNA can be performed with full notification, and prenatal diagnosis can be performed when necessary. Fetal aneuploidy (autosomal aneuploidy) screening based on cfDNA is by far the most superior screening method for Down Syndrome, with unprecedented sensitivity and specificity (99.7%) [39]. The detection rates of trisomy 18 and 13 and sex chromosome abnormalities (single X) were significantly lower than that of trisomy 21 (98.2%, 99.0%, and 95.8%, respectively) [39,40]. In the field of perinatal medicine, this technology is rapidly expanding to provide more and more clinical data. Some companies try to use cfDNA to provide screening conditions for microdeletion, such as 22q deletion and 1p36 deletion. However, these results are currently only used for prenatal screening, and it is not yet possible to make or exclude a prenatal diagnosis by this testing [24,41]. At present, many professional associations, including the SMFM and the ACOG, have issued recommendations for cfDNA screening that do not include the use of cfDNA in twin pregnancies. However, with the continuous research on cfDNA in multi-center clinical samples, it has been found that cfDNA detection is the most accurate screening method for trisomy 21 in twin pregnancies. Its screening performance is similar to that of a single pregnancy, and the failure rate is low. The prediction accuracy of trisomy 18 and trisomy 13 may be low. The author also believes that cfDNA screening for trisomy 21 is suitable for twin pregnancy, and we can further pay attention to the changes in the recommendations of professional associations.

Through the in-depth analysis of five hot topics of cfDNA in the field of obstetrics and gynecology from 2017 to 2021, we believe that as cfDNA technology becomes more sophisticated, it will have a broad role in prenatal screening, prenatal diagnosis (including preimplantation genetic testing), and pregnancy complications (such as pre-eclampsia and placental dysfunction). However, since cfDNA itself is obtained from maternal peripheral blood rather than directly from the fetus, it cannot replace invasive prenatal diagnosis technology, such as amniocentesis, in a short period of time.

Emerging frontiers in cfDNA research

Based on VOSviewer’s cfDNA frequent keyword–time dual-map and density map, we found that prenatal screening, genetic counseling and testing, fetal free DNA, and other topics have been studied earlier and are also in a leading position in research. The research density of cell-free fetal DNA, preimplantation genetic testing, etc., is relatively low. These topics have become research hotspots in recent years and have the potential for further research.

## 5. Limitation

Even though our paper provides a comprehensive review of the publications on cfDNA in obstetrics and gynecology from 2017 to 2021, similar to other bibliometric analyses, our paper has some limitations. First of all, our research only collected relevant articles from the PubMed and Web of Science databases. Although these databases contain a wealth of literature resources, there may still be some publications about recurrent miscarriages that have not been included. Second, there may be deviations in the author’s signature and the organization’s signature, resulting in a certain bias in the statistical results. Third, this study is based on similar existing cfDNA research so there are limitations in innovation, but we may provide inspiration for more innovative research in the future.

Overall, our analysis of the articles from 2017 to 2021 for the first time through a comprehensive visual analysis using co-authorship and co-occurrence methods identifies authors, organizations, countries/regions, journals, research hotspots, and emerging topics in the field of cfDNA for researchers interested in obstetrics and gynecology. We believe that this comprehensive study could make it easier for one to find a partner for project development and build a network of people who can help increase knowledge on this emerging topic.

## Figures and Tables

**Figure 1 jcm-11-05545-f001:**
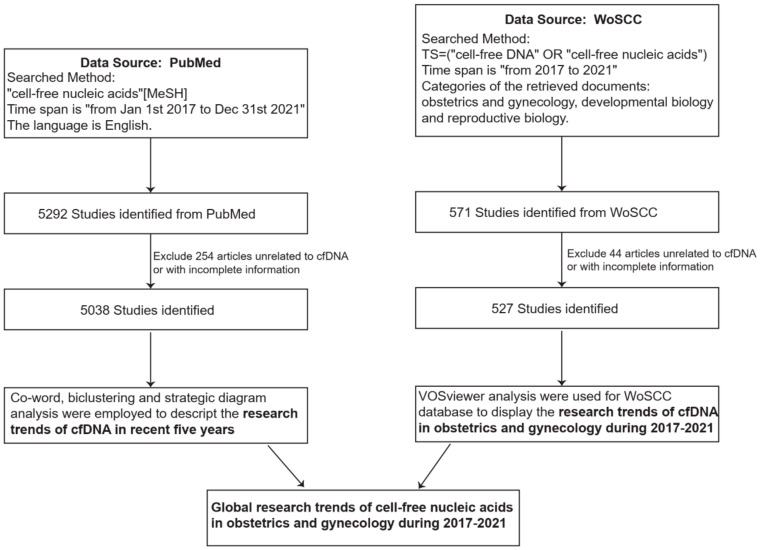
Data collection and analysis flow chart.

**Figure 2 jcm-11-05545-f002:**
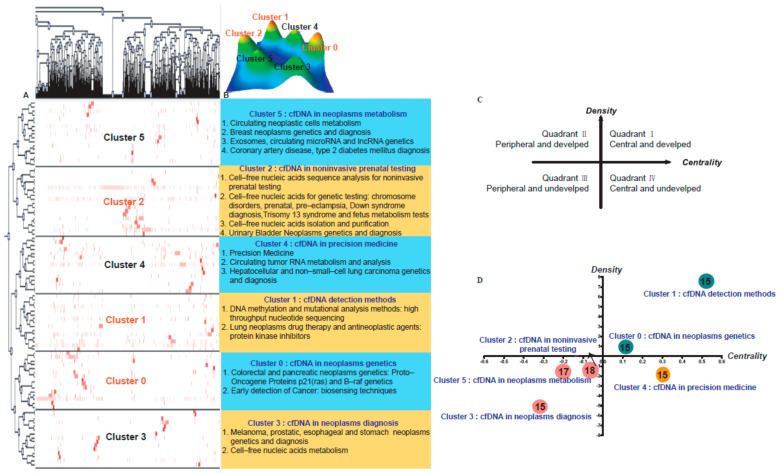
Biclustering analysis and strategy diagram of 95 high-frequency major MeSH subject/MeSH subheadings and articles on cfDNA downloaded from PubMed from 2017 to 2021. (**A**) Matrix visualization of biclustering of 95 high-frequency major MeSH subject/MeSH subheadings and PMIDs of articles. The number before each major MeSH subject/MeSH subheading represents the serial number as shown in Appendix A. (**B**) Mountain visualization of biclustering of 95 high-frequency major MeSH subject/MeSH subheadings and articles. (**C**) The significance of the four quadrants of the strategic diagram. (**D**) Clusters in each strategic diagram refer to the biclustering analysis results presented in Appendix A. The major MeSH terms/MeSH subheadings are represented by nodes in different quadrants. The number on the node represents the number of major MeSH terms/MeSH subheadings involved in each cluster. The arrows associated with clusters point to their descriptions. PMID: PubMed unique identifier.

**Figure 3 jcm-11-05545-f003:**
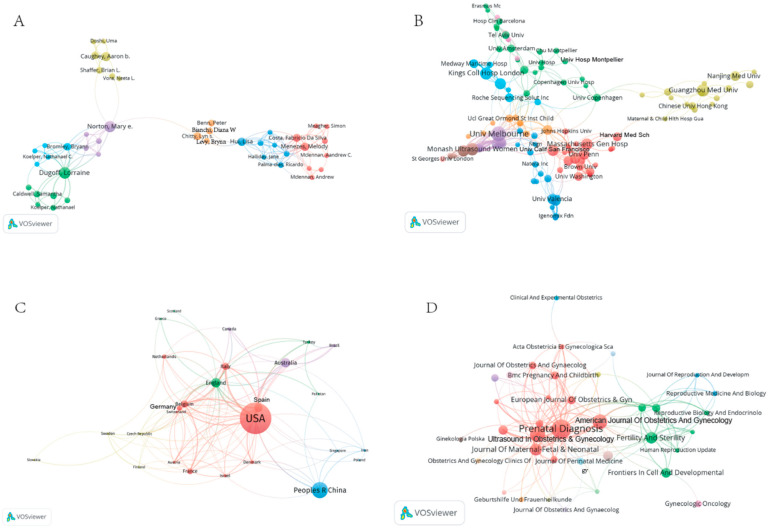
Distribution and clustering analysis of main research characteristics described by VOSviewer analysis in the cfDNA study from 2017 to 2021. (**A**) Distribution of the main first author in the cfDNA study from 2017 to 2021. (**B**) Distribution of the main research institutions in the cfDNA study from 2017 to 2021. (**C**) Distribution of the main research countries in the cfDNA study from 2017 to 2021. (**D**) Distribution of the main journals in the cfDNA study from 2017 to 2021. The color of a signal node is related to the category of each characteristic.

**Figure 4 jcm-11-05545-f004:**
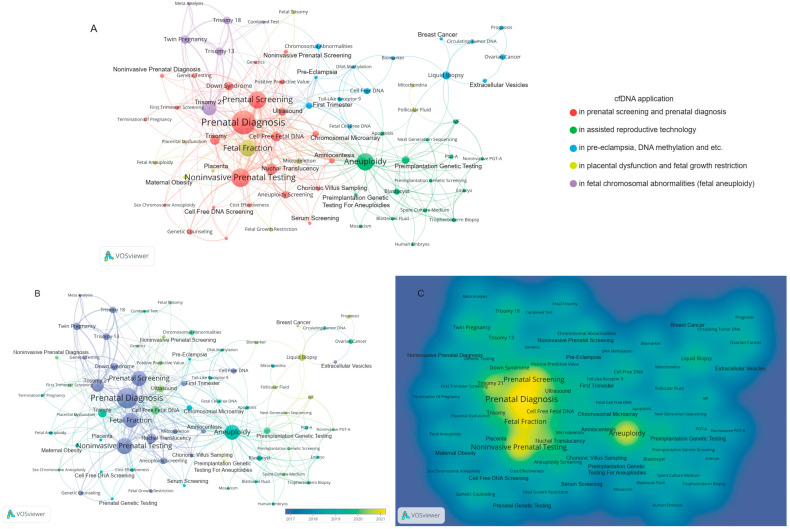
Keyword clustering analysis and co-occurrence networks of cfDNA in the articles related to the field of obstetrics and gynecology downloaded from WoSCC. (**A**) Keyword clustering analysis map of cfDNA in the field of obstetrics and gynecology based on VOSviewer. (**B**) Frequent keyword–time dual-map for frequent cfDNA-related keywords in the field of obstetrics and gynecology based on VOSviewer. (**C**) Density map of frequent cfDNA-related keywords in the field of obstetrics and gynecology based on VOSviewer. The color of a signal node is related to the category of each keyword cluster. WoSCC: Web of Science Core Collection.

**Figure 5 jcm-11-05545-f005:**
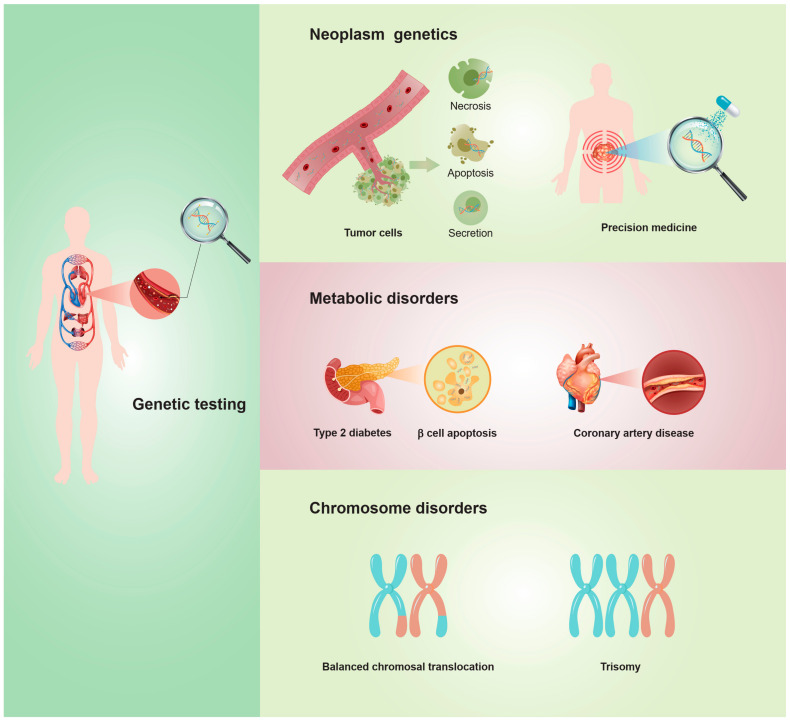
The main clinical applications of cfDNA genetic testing, including neoplasm genetics, metabolic disorders, and chromosomal disorders.

**Figure 6 jcm-11-05545-f006:**
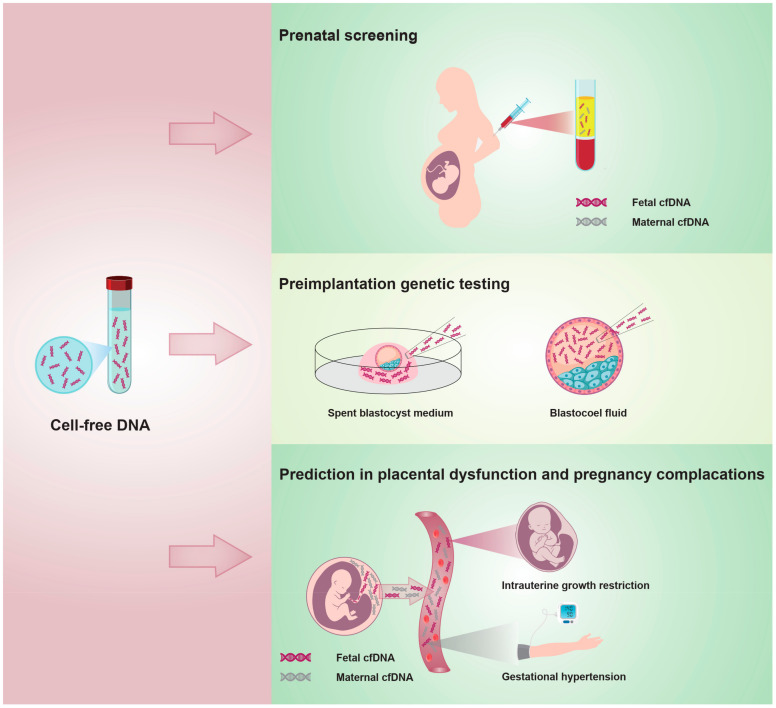
The application of cfDNA in the field of obstetrics and gynecology.

**Table 1 jcm-11-05545-t001:** Most influential authors and co-cited authors of cell-free DNA research. Related articles in obstetrics and gynecology from 2017 to 2021.

Rank	First Author	Articles	Co-Cited Author	Citations of Co-Cited Author
1	Dugoff, Lorraine	15	Levy, Brynn	119
2	Norton, Mary E	14	Norton, Mary E	74
3	Caughey, Aaron B	9	Hui, Lisa	68
4	Hui, Lisa	8	Dugoff, Lorraine	67
5	Menezes, Melody	8	Costa, Fabricio Da Silva	67
6	Chitty, Lyn S	7	Menezes, Melody	57
7	Costa, Fabricio Da Silva	7	Bianchi, Diana W	53
8	Avram, Carmen M	6	Mclennan, Andrew	51
9	Bromley, Bryann	6	Scott, Fergus	51
10	Benn, Peter	5	Nisbet, Debbie	50

**Table 2 jcm-11-05545-t002:** Most influential institutions that have published articles on cell-free DNA research in obstetrics and gynecology from 2017 to 2021.

Rank	Institutions	Countries	Number of Articles Published	Citations
1	Univ Melbourne	Australia	24	181
2	Guangzhou Med Univ	China	17	32
3	Kings Coll Hosp London	England	16	215
4	Monash Ultrasound Women	Australia	16	99
5	Monash Univ	Australia	16	116
6	Massachusetts Gen Hosp	USA	15	50
7	Murdoch Childrens Res Inst	Australia	15	121
8	Mercy Hosp Women	Australia	14	108
9	Univ Libre Bruxelles	Belgium	14	153
10	Univ Sydney	Australia	14	148

**Table 3 jcm-11-05545-t003:** Most influential countries with the number of published articles related to cell-free DNA research in obstetrics and gynecology from 2017 to 2021.

Rank	Countries	Articles	Citations
1	USA	189	1012
2	China	82	255
3	England	42	378
4	Australia	41	292
5	Spain	36	332
6	Belgium	29	240
7	Italy	29	226
8	France	24	62
9	Germany	19	64
10	Netherlands	16	134

**Table 4 jcm-11-05545-t004:** Most influential journals with the number of published articles related to cell-free DNA research on obstetrics and gynecology from 2017 to 2021.

Rank	Journal	Countries	Publications of Articles
1	Prenatal Diagnosis	England	90
2	Ultrasound in Obstetrics & Gynecology	England	51
3	American Journal of Obstetrics and Gynecology	USA	48
4	Journal of Maternal-Fetal & Neonatal Medicine	England	28
5	Fertility and Sterility	USA	27
6	Frontiers in Cell and Developmental Biology	Switzerland	17
7	Reproductive Sciences	USA	17
8	Fetal Diagnosis and Therapy	Switzerland	15
9	European Journal of Obstetrics & Gynecology and Reproductive Biology	Netherland	13
10	Human Reproduction	England	12

**Table 5 jcm-11-05545-t005:** Most influential keywords in the 5 categories in the cell-free DNA keyword co-occurrence analysis related to obstetrics and gynecology from 2017 to 2021.

I (Red)	II (Green)	III (Blue)	IV (Yellow)	V (Purple)
Prenatal Diagnosis	Aneuploidy	First Trimester	Fetal Fraction	Trisomy 21
Prenatal Screening	Preimplantation Genetic Testing (PGT)	Liquid Biopsy	Microdeletion	Trisomy 13
Non-invasive Prenatal Testing	Blastocyst	Cell-Free DNA	Fetal Trisomy	Twin Pregnancy
Nuchal Translucency	Spent Culture Medium	Chromosomal Abnormalities	Fetal Aneuploidy	Trisomy 18
Cell-Free Fetal DNA	Trophectoderm Biopsy	Pre-Eclampsia	Maternal Obesity	Meta-Analysis
Ultrasound	Blastocoel Fluid	Biomarker	Follicular Fluid	Combined Test
Amniocentesis	IVF	DNA Methylation	Fetal Growth Restriction	
Chorionic Villus Sampling	Preimplantation Genetic Screening	Fetal Cell-Free DNA	Placenta	
Chromosomal Microarray	Apoptosis	Circulating Tumor DNA	Mitochondria	
Genetic Testing	PGT-A	Toll-Like Receptor 9	Placental Dysfunction	

Red Cluster: cfDNA application in prenatal screening and prenatal diagnosis. Green Cluster: cfDNA application in assisted reproductive technology. Blue Cluster: cfDNA application in pre-eclampsia, etc. Yellow Cluster: cfDNA application in fetal growth restriction and placental dysfunction. Purple Cluster: cfDNA application in fetal chromosomal abnormalities (fetal aneuploidy).

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
