# Peer review of "Global Trends in Research on Cell-Free Nucleic Acids in Obstetrics and Gynecology during 2017–2021"

_jcm, 2022, doi:10.3390/jcm11195545_

Round 1
Reviewer 1 Report
The article under the title: “Global research trends of cell-free nucleic acids in obstetrics and gynecology during 2017-2021” represents an interesting attempt to summarize the global research trends of cfDNA investigation in the field of obstetrics and gynecology.
However, some major obstacles prevent the paper's publishing in its current form.
The whole paper looks more like the crude and unfinished draft version that has to be completely redesigned and rewritten more concisely and understandably.
The methodological parts of the manuscript are unnecessarily repeated a couple of times, and the discussion should be divided into several logical units. Also, the abstract of the manuscript should be rewritten more concisely.
The methodology diagram used for article selection (illustrating inclusion and exclusion criteria) should be incorporated into the manuscript.
Although the author's tendency and overall idea are highly appreciated, the manuscript should be elaborated.
Nevertheless, revision and resubmission of a manuscript are highly encouraged.
Reviewer 2 Report
· I would add in the keywords : prenatal screening
· Introduction:
o first line: I would change
Cell-free nucleic acids (cfNAs) are nucleic acids found in the cell-free parts
In
Cell-free nucleic acids (cfNAs) are nucleic acids found outside the cells/nucelo, in plasma and….
o is clarified that “Cell-free deoxyriboNucleic acid (cfDNA)” so there’s no need to repeat the interpretation in other tables/figures
o last sentence: is wrote too many times “topics”. Would be better to refer also to the “key world” as is done in the discussion
· Data source and research process:
o clarify that MeSH means Medical Subject Headings and don’t repeat it in any figures
o Report the total amount of paper that come out from the two online search.
· Figure 1
o put before A plot and then the B one, otherwise is not so well readable
o the text in the plot A is not readable at all. Better list only the name of the 6 clusters listed in table S2 and leave the possibility to go deeper in the high frequency terms on table S1
o Figure 1 and table S2: would be helpful give at each cluster a short name (key word) that clarify the topic and that use it the plot A and in the plot D
o Figure 1 description: is write that “size of signal node is related to the number of MeSH” but is not possible to appreciate the different size of each node, they appear all similar. Could be useful put number of papers for MeSH cluster
o Is not clear the methods used to define a cluster position in the plot. Clarify better in which case for you is well developed a cluster or not. Number of papers? Number of MeSH?
· Theme trends of cfDNA:
o clarify results of biclustering analysis to allow us the possibility to understand the assignment to the four quadrants strategic diagram.
o Clarify what do you mean for immature theme
· Table S3 and Figure2: error Bianchi,Ddiana W should be corrected in Bianchi,Diana W
· Figure 2
o Is not readable. Too much text in too less space, even if you zoom the PDF the text Is not readable. Clarify the colour legend
o Put in the paper table S3 that is more clear
· Most influential keywords in cfDNA research: you write that you retrieved 527 articles. Clarify how you get this number in Methods
· Table 1:
o Give a name to the 5 categories to clarify why you clustered them using the description
§ prenatal screening and prenatal diagnosis
§ assisted reproductive technology
§ pre-eclampsia, DNA methylation and etc.
§ placental dysfunction and fetal growth restriction
§ fetal chromosomal abnormalities (fetal aneuploidy)
o Table it with visible excel grid, otherwise the wrap text is not readable
o In the discussion you talk about niPGT many times, but is not listed in the II green category
· Figure 3:
o The scope of plot B and C Is not clear and hardly readable. List some key world and show in 5 years if publication (%) increase or decrease if the scope is to show that some hot topics decreased in the last five years and some other are increasing.
o Add color legend with 5 categories description
· Figure 4:
o Change “The main clinical applications of genetic testing, including neoplasms…” with “The main clinical applications of cfDNA genetic testing, including neoplasms…”
o The first picture on the left show a men with a cell with a nucleo and inside DNA, but the paper discuss about circulating DNA and probably the picture is misleading
· Discussion:
o From line 81 you discuss a lot about each of your 5 key word categories, but in my opinion this is off topics because you’re not writing a review but discussing about key world mostly used in publication and consequently on research. For sure you can write wat’s going on in the 5 topics, but you could better conclude why some topics are increasing or not, why research is moved form basic screening to preimplantation or vice versa thanks to your world wide view.
o Line 91 to 94: clarify that amniocentesis analysis still be the only one method to see balanced rearrangement and mosaicism.
o Move line 170 to 182 in the green cluster discussion part
o Line 186: you name thalassemia as a key world but I could not find it in Fig 3. Check it
o Line 194: change “recurrent abortion” in “recurrent miscarriage”
o Line 201 and 202: could be better clarified that thanks to this comprehensive study could be easier find partner for project development and build a network of person who can lead the increase of knowledge in this emerging topic
· Bibliography 9:
o Add also 2022 update of the same paper: Roskoski R Jr. Properties of FDA-approved small molecule protein kinase inhibitors: A 2022 update. Pharmacol Res. 2022 Jan;175:106037.
Round 2
Reviewer 2 Report
text changes are fine to me and the paper seem really well organized and clear now.
unfortunately I was not able to check figures because they was not reported in the PDF or in the supplementary file. check the resolution before sending last version
Author Response
Thank you for your recognition and reminder. In the last file submission, we uploaded the pictures and supplementary table as attachments. It could be our fault that something went wrong. Very sorry for our mistakes. This time, we uploaded all the pictures and supplementary tables in the revised PDF manuscript, and the picture clarity of the PDF version is very good.